# Anticoagulation Strategies for Left Ventricular Thrombus After Myocardial Infarction: A Review

**DOI:** 10.3390/jcm14227982

**Published:** 2025-11-11

**Authors:** Adam Folman, Nicola Toukan, Ofer M. Kobo, Ariel Roguin, Maguli S. Barel

**Affiliations:** 1Division of Cardiovascular Medicine, Hillel Yaffe Medical Center, Hadera 3820302, Israelaroguin@technion.ac.il (A.R.); 2The Ruth and Bruce Rappaport Faculty of Medicine, Technion—Israel Institute of Technology, Haifa 3200003, Israel

**Keywords:** anticoagulation, complications, left ventricle, myocardial infarction, outcomes, thrombus

## Abstract

Left ventricular thrombus (LVT) remains a clinically significant complication following acute myocardial infarction (MI). Although its incidence has declined in the era of primary percutaneous coronary interventions (PCIs), the best treatment remains unclear. For decades, vitamin K antagonists (VKAs) such as warfarin have been the mainstay of therapy, supported by guidelines recommendations. However, the limitations of warfarin, including a narrow therapeutic range, the need for frequent monitoring, and food/drug interactions, have spurred interest in direct oral anticoagulants (DOACs). This review summarizes the available evidence on anticoagulation strategies for LVT after MI, focusing on observational studies and recent randomized controlled trials. A total of 12 studies were included in this review: 9 retrospective cohorts and 3 randomized controlled trials. Patient populations ranged from small single-center cohorts to large multicenter registries. DOACs, compared with warfarin, were associated with a higher rate of thrombus resolution, a lower rate of stroke and systemic embolism, and a similar mortality. The usage of DOACs marginally reduced the rate of major bleeding compared with warfarin. The current evidence indicates that DOACs may offer comparable efficacy and potentially improved safety relative to warfarin, although most randomized trials remain small and underpowered for definitive conclusions. Larger, adequately powered studies are still required before DOACs can be routinely considered equivalent alternatives. The RIVAWAR randomized trial provides the strongest evidence to date regarding the use of DOACs in LVT after MI, but further large-scale randomized studies are required to establish definitive guidance. Until then, anticoagulation therapy including DOACs should be individualized, balancing the thromboembolic risk, bleeding risk, and practical considerations of anticoagulant use.

## 1. Introduction

LVT is a well-recognized complication of acute MI. Although its incidence has markedly declined in the contemporary reperfusion era, its clinical impact remains substantial, with risks of systemic embolization, stroke, and increased mortality.

Historically, before the advent of reperfusion, LVT was observed in 30–40% of patients with large anterior infarcts, particularly those with a severely reduced left ventricular ejection fraction (LVEF) [1]. With the introduction of thrombolytic therapy, the incidence was reduced to 10–20% [2]. In the primary percutaneous coronary intervention (PCI) era, the prevalence is typically reported as 2–6% using standard transthoracic echocardiography (TTE) but rises to 15–20% when assessed by cardiac magnetic resonance (CMR), which is more sensitive to small and mural thrombi [3,4]. These findings highlight both the success of reperfusion strategies and the ongoing underdiagnosis of LVT in routine practice.

The clinical consequences of LVT are serious. Systemic embolism, most often ischemic stroke, occurs in 10–15% of untreated patients within the first 3 months following MI [5]. Beyond embolic events, LVT has been linked to nearly double the risk of long-term mortality compared with patients without thrombi [4]. Predictors of thrombus formation include large anterior infarcts, apical akinesis or dyskinesis, and persistent LV systolic dysfunction [2,6]. These high-risk subgroups emphasize the importance of careful surveillance and the timely initiation of anticoagulation.

The pathogenesis of LVT can be conceptualized through Virchow’s triad. First, blood stasis develops within akinetic or dyskinetic myocardial segments, particularly in the apex after anterior MI. Second, endothelial injury caused by ischemic necrosis and tissue remodeling creates a thrombogenic surface. Third, systemic hypercoagulability in the post-MI inflammatory state further amplifies clot formation [6]. These mechanisms provide a strong biological rationale for anticoagulation, aiming to both prevent embolization and promote thrombus resolution.

VKAs, primarily warfarin, have historically been the treatment of choice. Their efficacy in reducing embolic risk has been supported by decades of clinical experience and guideline endorsement [7,8]. However, VKAs suffer from well-known limitations: a narrow therapeutic window, a delayed onset of action, frequent INR monitoring requirements, and numerous food–drug and drug–drug interactions [9]. In real-world practice, these drawbacks often lead to suboptimal anticoagulation control, poor adherence, and higher rates of both embolic and bleeding complications [10].

The introduction of DOACs—including rivaroxaban, apixaban, dabigatran, and edoxaban—has revolutionized anticoagulation in atrial fibrillation (AF) and venous thromboembolism (VTE). Large randomized trials have consistently shown that in AF, DOACs are at least as effective as warfarin, with superior safety regarding intracranial hemorrhage [11,12,13,14]. Their advantages include predictable pharmacokinetics, fixed dosing, a rapid onset of action, and a reduced need for monitoring, making them appealing alternatives in off-label scenarios such as LVT.

Early retrospective studies of DOACs in LVT suggested a non-inferior efficacy compared with warfarin, with potential reductions in bleeding risk [11]. More recently, small randomized trials have provided prospective evidence supporting apixaban and rivaroxaban as viable alternatives [12,13]. The landmark RIVAWAR trial, published in 2025, further confirmed the non-inferiority of rivaroxaban compared with warfarin for thrombus resolution and embolic prevention, with fewer clinically relevant bleeding events [14].

Despite these advances, international guidelines remain conservative. Both the 2017 ESC STEMI guidelines and the 2013 ACC/AHA STEMI guidelines continue to recommend VKAs for LVT treatment [7,8]. Nonetheless, the growing amount of real-world data and emerging randomized evidence is shifting the paradigm. In daily clinical practice, many clinicians now consider DOACs for patients in whom warfarin is impractical, such as those with poor INR stability or limited access to monitoring facilities.

In this review, we synthesize the evidence comparing DOACs and VKAs for the management of LVT following MI. We present detailed study-level findings; highlight outcomes including thrombus resolution, embolic complications, bleeding, and mortality; and contextualize these results within the framework of current guidelines. Finally, we outline key knowledge gaps and priorities for future research, including the optimal treatment duration, standardized imaging follow-up, and management strategies for high-risk patient subgroups.

## 2. Methods

This article was conducted as a review of the literature examining anticoagulation strategies for LVT after MI.

We chose a narrative synthesis rather than a quantitative meta-analysis, given the marked heterogeneity of the available evidence with respect to the study design, patient populations, imaging modalities, treatment protocols, and reported outcomes.

### 2.1. Search Strategy and Study Selection

A systematic literature search was performed in PubMed, Embase, and the Cochrane Library from database inception to August 2025. The following search terms were used in various combinations: “left ventricular thrombus”, “myocardial infarction”, “warfarin”, “vitamin K antagonist”, “direct oral anticoagulant”, “DOAC”, and “NOAC”. Reference lists of relevant reviews and included studies were hand-searched to identify additional eligible publications [15] [Figure 1].

Studies were considered eligible if they met all of the following criteria:Enrolled adult patients (≥18 years) with documented LVT following MI, confirmed by echocardiography, contrast echocardiography, or cardiac magnetic resonance (CMR);Included a comparison between a direct oral anticoagulant (DOAC: rivaroxaban, apixaban, dabigatran, or edoxaban) and a vitamin K antagonist (VKA: warfarin);Reported at least one relevant outcome, including thrombus resolution, systemic embolism or ischemic stroke, major bleeding, or all-cause mortality.

We excluded case reports, case series with less than 10 patients, review articles, editorials, and studies that did not include a comparator arm. When overlapping cohorts were reported, the study with the largest sample size or most complete dataset was retained.

### 2.2. Data Extraction and Outcomes of Interest

Data extraction was performed independently by two authors. Extracted variables included study design (retrospective, prospective, or randomized controlled trial), geographic region, sample size, baseline patient characteristics, method of thrombus detection, anticoagulation regimen and duration, concomitant antiplatelet therapy, imaging follow-up protocols, and study outcomes.

The primary efficacy outcome was **thrombus resolution**, assessed by follow-up imaging at 1–12 months. Secondary outcomes included **systemic embolism or ischemic stroke**, **major bleeding** (as defined by each study, including ISTH, TIMI, or BARC criteria), and **all-cause mortality**.

Additional exploratory outcomes, such as recurrent MI, rehospitalization, or recurrence of thrombus, were recorded when available.

All pooled risk ratios reported in this review are derived from previously published analyses and should not be interpreted as results of an original pooled analysis performed.

### 2.3. Quality Assessment

Given the predominance of observational data in this field, we assessed study quality using the Newcastle–Ottawa Scale (NOS) for cohort studies [16]. Domains assessed included selection of study groups, comparability of cohorts, and outcome assessment. For randomized controlled trials (RCTs), the Cochrane Risk of Bias 2 tool was applied [17]. Particular attention was paid to randomization processes, allocation concealment, blinding, completeness of outcome reporting, and potential conflicts of interest.

### 2.4. Synthesis of Evidence

Due to significant heterogeneity in study design, populations, imaging methods, and definitions of outcomes, a formal quantitative meta-analysis was not performed. Instead, we employed a structured narrative synthesis, highlighting both individual study results and pooled estimates when available from recent meta-analyses [2,18]. Subgroup observations, such as differences by infarct location (anterior vs. non-anterior MI) and imaging modality (TTE vs. CMR), were also noted when reported.

Our aim was to provide a comprehensive overview of the evolving evidence, integrating randomized and observational data with mechanistic insights and guideline perspectives, to inform clinical decision-making and identify priorities for future research.

### 2.5. Used of GenAI

During the preparation of this manuscript, the author used ChatGPT (OpenAI, GPT-5) for language editing and paragraph restructuring during manuscript drafting. The author has reviewed and edited the output and takes full responsibility for the content of this publication.

## 3. Results

A total of 12 studies were included in this review: 9 retrospective cohorts and 3 randomized controlled trials. Patient populations ranged from small single-center cohorts to very large multicenter registries. Key study characteristics and clinical outcomes are summarized in Table 1, Table 2 and Table 3.

### 3.1. Thrombus Resolution

There were 11 studies involving 886 participants that reported thrombus resolution. DOACs were associated with a higher rate of thrombus resolution compared with warfarin, with a pooled RR of 1.02 (95% CI 0.98–1.06) according to those studies [8,9,10,11,12].

In randomized controlled trials, no statistical significance for thrombus resolution was observed between DOACs and warfarin, with an RR of 0.99 (95% CI 0.89–1.10). The subgroup analysis stratified by the enrolled study population, including STEMI-only cohorts and those enrolling all MI types, demonstrated consistent findings. A further analysis of thrombus resolution at 1 year, which included results from four studies involving 320 participants, showed a consistent efficacy of DOACs compared to warfarin.

### 3.2. Stroke and Systemic Embolism

There were nine studies involving 14,981 participants that reported data on stroke and systemic embolisms. DOACs were associated with a lower rate of stroke and systemic embolism, with a pooled RR of 0.84 (95% CI 0.78–0.90) [8,9,10,11,12].

### 3.3. Major Bleeding

There were 10 studies involving 15,123 participants that reported data on major bleeding [8,9,10,11,12]. The definition of major bleeding varied among the studies and is presented in Table 3. DOACs marginally reduced the rate of major bleeding compared with warfarin, with a pooled RR of 0.87 (95% CI 0.76–1.01).

### 3.4. All-Cause Mortality

There were seven studies involving 548 participants that reported all-cause mortality. No significant difference was observed between DOACs and warfarin, with a pooled RR of 1.08 (95% CI 0.98–1.19) [8,9,10,11,12].

## 4. Discussion

This review highlights several key insights into anticoagulation strategies for LVT after MI. First, across 12 studies, DOACs appear comparable—but definitively not superior to warfarin in promoting thrombus resolution, with resolution rates generally between 70 and 85% with both treatment options. Second, embolic event rates were low and not significantly different between groups, although observational heterogeneity limits firm conclusions. Third, DOACs may confer a modest safety advantage, with numerically fewer major bleeding events in several studies.

From a mechanistic perspective, the pathophysiology of LVT formation relates to stasis in akinetic or dyskinetic myocardial segments, endothelial injury from infarction, and hypercoagulability [6,9]. Anticoagulation is therefore a logical therapeutic cornerstone [18]. The choice between DOACs and warfarin must also consider real-world factors: DOACs offer convenience and fewer interactions, while warfarin remains the most extensively studied option and the only guideline-endorsed therapy [4,5].

The RIVAWAR trial provides the first randomized evidence in this space, showing the non-inferiority of rivaroxaban compared to warfarin [14]. This was a single-center, open-label, non-inferiority, randomized, and controlled trial involving patients with acute LVT diagnosed during their initial MI hospitalization. Participants were randomized 2:1 to receive either rivaroxaban (20 mg daily) or warfarin (target international normalized ratio 2–3) for 12 weeks. The primary endpoint, thrombus resolution, was assessed by echocardiography at 4 and 12 weeks. A total of 261 patients were randomized, with 171 in the rivaroxaban group and 90 in the warfarin group. The groups were similar in sex, age, and MI subtype, with most having ST-segment elevation MI and severe LV dysfunction. At 4 weeks, the thrombus resolution rate was higher in the rivaroxaban group (20% vs. 8%; *p* = 0.017), with a similar resolution at 12 weeks (95.8% vs. 96.6%; *p* = 0.759). The cumulative all-cause mortality was comparable (3.5% vs. 3.3%; *p* = 0.921). Major bleeding occurred in 2.3% of rivaroxaban patients vs. 1.1% of the warfarin group (*p* = 0.491). The authors concluded that rivaroxaban demonstrated a similar efficacy to warfarin in treating post-MI LVT, with >95% resolution in both groups. While encouraging, RIVAWAR should be interpreted as hypothesis-generating rather than practice-changing. Larger multicenter studies with longer follow-up are essential before DOACs can be considered equivalent to VKAs in all settings.

The biological rationale for anticoagulation in LVT is firmly rooted in Virchow’s triad: the stasis of blood in akinetic or dyskinetic myocardial regions, endothelial injury from infarction, and hypercoagulability in the post-MI state [1]. Patients with large anterior infarctions and apical akinesis are at a particularly high risk, as there is more stasis in the akinetic areas of the LV [2] [Figure 2].

VKAs inhibit the vitamin K–dependent synthesis of clotting factors II, VII, IX, and X. Their efficacy is well established but limited by their delayed onset, narrow therapeutic window, genetic variability in metabolism, and multiple food–drug interactions [8]. DOACs, in contrast, provide direct factor Xa or thrombin inhibition with predictable pharmacokinetics and a rapid onset [9,18]. These features make them theoretically attractive in the early post-MI setting, when the embolic risk is highest.

However, not all is promising for the clinical use of DOACs. The experience from other thrombotic conditions highlights limitations. DOACs failed in large studies with mechanical valves. The RE-ALIGN trial of dabigatran in mechanical prosthetic valves was terminated early due to excess thromboembolic and bleeding events [25]. Similarly, the [26] Xa trial evaluating apixaban in mechanical aortic valves was halted for futility [27]. Apixaban failed to meet non-inferiority criteria and showed a higher rate of blood clots and strokes than warfarin. Therefore, the guidelines explicitly advise against DOAC use in patients with mechanical valves. In these patients, warfarin remains the gold standard. These failures underscore that not all thrombotic environments are equivalent. LVT develops in low-flow conditions, physiologically closer to venous thrombosis, where DOACs have demonstrated robust efficacy, supporting their potential role in this context.

Guidelines have historically favored VKAs. The 2023 ESC Guidelines for the Management of Acute Coronary Syndromes provide the first unified guidance spanning STEMI and NSTE-ACS and emphasize long-term antithrombotic management [6,7]. While these guidelines do not specify an optimal anticoagulant agent for patients with left ventricular thrombus (LVT), they acknowledge that oral anticoagulant therapy (vitamin K antagonists or direct oral anticoagulants) may be considered in patients with confirmed LVT (Class IIa, Level of Evidence C). Importantly, the document reinforces the need for individualized decisions on the duration of therapy and combination with antiplatelet agents according to the ischemic and bleeding risk. Incorporating these newer recommendations provides s context for our review and underscores the gap in high-quality evidence specific to LVT management. Similarly, the 2013 ACCF/AHA STEMI guideline endorses VKAs [8]. These positions are based largely on observational evidence accumulated over decades, before DOACs were available.

More flexible positions have emerged in recent years. Canadian guidance allows DOAC use when VKAs are impractical [28], and Japanese guidelines also acknowledge their use in selected patients [29]. The publication of prospective RCTs—Alcalai et al. (2022, apixaban vs. warfarin) [12], Youssef et al. (2023, apixaban vs. warfarin) [13], and the landmark RIVAWAR trial (2025, rivaroxaban vs. warfarin) [14]—has provided new evidence that DOACs are non-inferior to warfarin for thrombus resolution and embolic prevention, with favorable safety profiles. While these trials remain small compared with AF or VTE studies, they are likely to influence future guideline updates.

**Imaging and Monitoring:** The accurate detection and monitoring of LVT are essential for guiding therapy. Most studies employed transthoracic echocardiography (TTE), which, while accessible and inexpensive, underestimates the thrombus prevalence [3]. Contrast echocardiography increases sensitivity but remains limited compared with cardiac magnetic resonance (CMR), which is the gold standard [3,4].

Studies using CMR consistently demonstrate a higher baseline thrombus prevalence and lower apparent resolution rates, reflecting its ability to detect small or mural thrombi missed by TTE [3]. This discrepancy complicates the comparison of treatment outcomes across studies.

Guidelines recommend repeat imaging at 3 months, but the optimal timing is uncertain. In patients with persistent severe LV dysfunction, the thrombus may recur despite an initial resolution [4]. Standardized imaging protocols and follow-up intervals are needed to refine the treatment duration and improve comparability across future studies. Conversely, some patients with small thrombi and rapid LV recovery may not require extended therapy. There are no guidelines for small thrombi with rapid LV recovery, so decisions are individualized and based on imaging and the clinical trajectory. The risk of embolism depends not only on the thrombus size but also on the mobility, location, and underlying LV dysfunction. In each patient, the bleeding risk must be weighed, especially in patients with comorbidities or those at higher hemorrhagic risk.

**Concomitant Antiplatelet Therapy:** One of the most challenging clinical issues in LVT management is balancing anticoagulation with antiplatelet therapy. The majority of post-MI patients undergo PCI and require dual antiplatelet therapy (DAPT). When combined with oral anticoagulation, this results in triple therapy, which markedly increases the bleeding risk [5].

Evidence from atrial fibrillation–PCI populations provides guidance. The PIONEER AF-PCI trial (rivaroxaban) [30], RE-DUAL PCI trial (dabigatran) [31], and AUGUSTUS trial (apixaban) [32] all showed that dual therapy (OAC + single antiplatelet) reduces bleeding compared with triple therapy, without increasing ischemic events. Although extrapolation to LVT should be cautious, many clinicians apply a pragmatic strategy of 1 month of triple therapy followed by OAC plus a single antiplatelet agent.

In this context, DOACs may offer an advantage. Compared with VKAs, they are associated with lower rates of intracranial bleeding [9], which is particularly relevant when combined with antiplatelets. Observational studies in LVT populations also suggest numerically fewer bleeding complications with DOAC-based regimens [11,15].

In clinical practice, the optimal regimen should therefore be individualized by weighing embolic and bleeding risks. Patients with large anterior infarctions or a severely reduced LVEF may benefit from a longer duration of anticoagulation, whereas those with a high bleeding risk or recent PCI often require an earlier de-escalation to dual therapy. Risk assessment tools such as HAS-BLED or PRECISE-DAPT can assist in estimating the bleeding risk, while imaging follow-up at 3 months allows for the reassessment of the embolic risk once thrombus resolution is confirmed. This pragmatic, patient-centered approach reflects the nuanced decision-making required in managing LVT after MI rather than a uniform protocol for all patients.

Several population subgroups warrant tailored consideration:[1]**Persistent LV dysfunction:** Patients with ongoing severe systolic dysfunction remain at high risk for recurrent thrombi, with recurrence rates up to 15% [4]. Extended anticoagulation beyond 6 months may be appropriate, but the optimal duration is unknown.[2]**Elderly patients:** Older age increases both thrombotic and bleeding risks. DOACs may be preferable due to reduced intracranial bleeding [9], but renal clearance must be carefully monitored.[3]**Chronic kidney disease (CKD):** VKAs can be used across the full spectrum of renal functions, while DOAC dosing must be adjusted or avoided in advanced CKD. Observational studies of DOACs in CKD populations show reassuring efficacy and safety, though data in LVT are sparse [27].[4]**Non-ischemic cardiomyopathy:** LVT also occurs in dilated or hypertrophic cardiomyopathy. Small case series suggest DOACs may be effective, but evidence is limited, and further study is required [33].

Beyond efficacy and safety, practical considerations strongly influence anticoagulant choices. Recent evidence has begun to explore the pharmacoeconomic and patient-centered implications of DOAC therapy [34,35,36,37,38,39,40]. Cost-effectiveness analyses indicate that, despite higher acquisition costs, DOACs may offer favorable value by reducing monitoring visits, adverse events, and hospitalizations compared with warfarin [37]. In parallel, several studies show that patients on DOACs report higher treatment satisfaction, better adherence, fewer lifestyle constraints, and improved quality of life compared with those on warfarin [36,37,38,39]. These findings reinforce that anticoagulant selection should integrate not only efficacy and safety but also economic and patient experience dimensions. In real-world practice, many clinicians already prescribe DOACs for LVT, particularly when INR instability or patient preference make warfarin unattractive [11]. This highlights a disconnect between formal guidelines and actual practice, reflecting both evolving evidence and patient-centered considerations.

Furthermore, real-world safety data reinforce the clinical applicability of DOACs. A recent multicenter study [40] reported that DOAC use in post-MI and cardiomyopathic populations was associated with low major bleeding rates and favorable net clinical benefits compared with VKAs. Such evidence strengthens the confidence in the safety profile of DOACs beyond controlled trial settings.

From a clinical standpoint, the accumulating evidence suggests that DOACs may be considered as an alternative to warfarin for selected patients with LVT after MI. DOACs are particularly attractive in situations where INR monitoring is challenging, patient adherence to frequent testing is problematic, or drug–food interactions with warfarin pose significant management difficulties. Their fixed dosing and favorable safety profile in atrial fibrillation and venous thromboembolism lend further support.

Nevertheless, caution is warranted. Warfarin remains the only agent explicitly endorsed by current guidelines, and long-term data on LVT populations are limited. DOACs should therefore be considered on an individualized basis, particularly in patients without contraindications, without mechanical valves, and with anticipated short-to-intermediate treatment courses.

Our review also emphasizes knowledge gaps. The optimal duration of anticoagulation, particularly in patients with persistent LV dysfunction, remains uncertain. Furthermore, the interplay between anticoagulation and concomitant dual antiplatelet therapy requires an individualized balancing of ischemic versus bleeding risks [10,11,12,13,14]. Imaging follow-up strategies (echocardiography vs. cardiac MRI) should be standardized to better compare outcomes across studies. Several steps are desirable. First, larger RCTs are urgently needed, focused on firm endpoints such as embolic events and mortality, rather than surrogate outcomes like thrombus resolution. Second, head-to-head comparisons of different DOACs are lacking; most studies have focused on apixaban and rivaroxaban. Third, standardized imaging protocols should be adopted across studies to ensure consistency. Fourth, the optimal duration of anticoagulation in patients with persistent LV dysfunction must be defined. Fifth, registry-based pragmatic trials could provide real-world data on recurrence, cost-effectiveness, and long-term safety.

Beyond pharmacologic anticoagulation, several non-anticoagulant strategies play a key role in preventing LVT formation and subsequent cardioembolic events. Early and complete myocardial reperfusion through primary PCI or fibrinolysis remains the most effective measure to limit the infarct size and apical akinesis, thereby reducing the substrate for thrombus formation. Optimized left ventricular unloading and remodeling prevention with evidence-based medical therapy—including ACE inhibitors, β-blockers, and mineralocorticoid receptor antagonists—further reduce stasis and endothelial injury. Additionally, antiplatelet therapy optimization, tight control of inflammatory and metabolic risk factors, and structured cardiac rehabilitation contribute to lowering the thrombotic potential. Emerging approaches targeting myocardial inflammation and prothrombotic signaling, as reviewed by Sgarra et al. [27], illustrate how comprehensive post-MI management may complement anticoagulation in mitigating embolic risk.

Finally, emerging technologies such as artificial intelligence-assisted imaging and biomarker-driven risk stratification hold promise for the earlier identification of patients at the highest risk of thrombus formation or embolization [33]. Such tools could refine patient selection for therapy and optimize follow-up strategies.

## 5. Conclusions

This review consolidates the available evidence comparing VKAs and DOACs in the treatment of LVT following MI. The available evidence—primarily from observational cohorts and small randomized trials—suggests that DOACs may be comparable to warfarin in achieving thrombus resolution and preventing embolic events, with a possible reduction in major bleeding. However, most studies are limited by modest sample sizes, short follow-ups, and the absence of firm clinical endpoints such as stroke or mortality. While these findings are encouraging, they should be interpreted with caution. The randomized data to date are insufficient to establish the non-inferiority or superiority of DOACs for all patient subgroups. Until large, adequately powered trials are completed, VKAs remain the reference standard, and DOACs should be considered selectively—particularly when INR control is unstable or contraindications to VKAs exist.

## Figures and Tables

**Figure 1 jcm-14-07982-f001:**
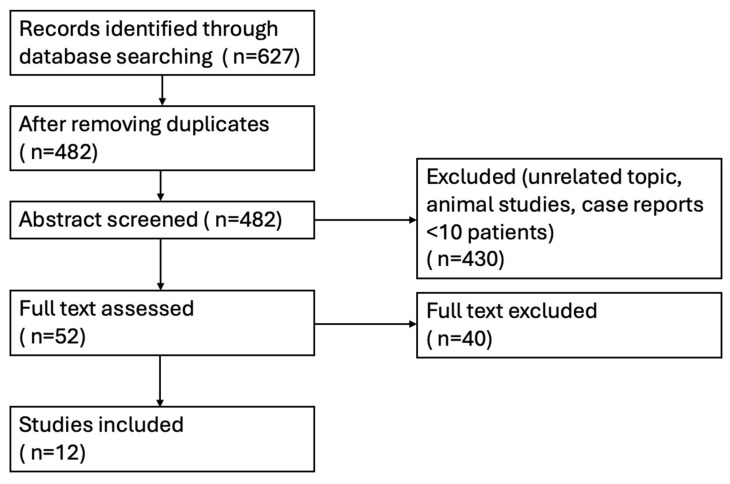
Flow diagram for paper screening.

**Figure 2 jcm-14-07982-f002:**
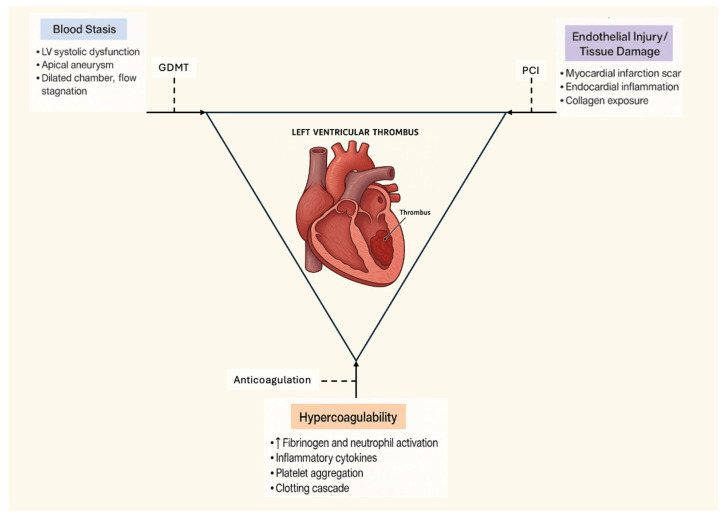
Flow diagram for LV thrombus formation and management.

**Table 1 jcm-14-07982-t001:** Characteristics of studies included in this review.

Study, Year (Country)	Design/ Population	LV Thrombus Detection	DOAC Regimen (n)	Warfarin Regimen (n/INR)	Antiplatelet/ Duration	Follow-Up
Albabtain, 2021 [19] (Saudi Arabia)	Retrospective; LVT post-MI	TTE	Rivaroxaban (n = 6)	n = 7, INR 2–3	Anticoag median 9.5 mo (6–32.5)	Median 36 mo
Alcalai, 2022 [12] (Israel)	RCT; LVT 1–14 days post-MI	TTE	Apixaban (n = 17)	n = 15, INR 2–3	A + C (1 mo) → C; 3 mo	3 mo
Jones, 2021 [15] (UK)	Prospective; post-MI with PCI	Echo or CMR	Riva 57.9%, Apix 36.8%, Edox 5.3% (n = 38)	n = 60 (INR NR)	NA	Median 1.8 y
Byrne, 2023 [6] (UK)	Retrospective; post-MI	NA	Riva 62.5%, Apix 31.3%, Edox 6.3% (n = 16)	n = 41 (INR NR)	SAPT 12.5%/DAPT 88% vs. SAPT 7.5%/DAPT 90%	1 y
Jaidka, 2018 [20] (UK)	Retrospective; anterior-wall STEMI	TTE; contrast echo (83.3%)	n = 9	n = 21 (INR NR)	6 mo	6 mo
Jones, 2021 [15] (UK)	Prospective; post-STEMI	TTE; contrast 22.8%; CMR 70.3%	Riva 58.5%, Apix 36.5%, Edox 5% (n = 41)	n = 60, INR 2–3	DAPT 68.3%/SAPT 24.4%; triple therapy median 3 mo	Median 2.2 y
Liang, 2022 [21] (China)	Retrospective; anterior-wall STEMI	TTE	n = 56 (Riva 48, Dabig 8)	n = 72, INR 2–3	Mostly triple (A + C/T); imaging at 3, 6, 12 mo	12 mo
Mansouri, 2024 [22] (Iran)	RCT; LVT in ACS	TTE	Rivaroxaban (n = 26)	n = 26 (INR NR)	NA	3 mo
Yao, 2025 [23] (International)	Retrospective; LVT in ACS	NA	n = 7151	n = 7151	NA	90 days
Youssef, 2022 [13] (Saudi Arabia)	RCT; post-anterior MI	TTE	Apixaban (n = 25)	n = 25, INR 2–3	DAPT 1 mo 80%; A 80%; C 96–100%; ≥3 mo	6 mo
Zhang, 2022 [24] (China)	Retrospective; post-STEMI	TTE	n = 33	n = 31, INR 2–2.5	A + C median 8.5 mo; triple median 8.5 mo (5–17)	Median 25 mo
RIVAWAR, 2025 [14] (International)	RCT; post-MI LVT	TTE	Rivaroxaban (n ≈ 130)	Warfarin (n ≈ 131), INR 2–3	DAPT permitted; 3 mo	3 mo

ACS—acute coronary syndrome, A—aspirin, A + C—aspirin plus clopidogrel, C—clopidogrel, CMR—cardiac magnetic resonance imaging, DAPT—dual antiplatelet therapy, DOAC—direct oral anticoagulant, Edox—edoxaban, Echo—echocardiography, INR—international normalized ratio, LVT—left ventricular thrombus, MI—myocardial infarction, mo—month(s), NA—not available/not applicable, NR—not reported, PCI—percutaneous coronary intervention, RCT—randomized controlled trial, Riva—rivaroxaban, Apix—apixaban, Dabig—dabigatran, SAPT—single antiplatelet therapy, STEMI—ST-segment elevation myocardial infarction, TTE—transthoracic echocardiography, and y—year(s).

**Table 2 jcm-14-07982-t002:** Reported outcomes across studies.

Outcome	Studies/N	Pooled Effect (DOACs vs. Warfarin)
Thrombus resolution	10 studies/625 pts	RR 1.02 (0.98–1.06)
Stroke/systemic embolism	8 studies/14,720 pts	RR 0.84 (0.78–0.90)
Major bleeding	9 studies/14,862 pts	RR 0.87 (0.76–1.01)
Any bleeding	7 studies/443 pts	RR 0.57 (0.33–1.00)
All-cause mortality	6 studies/287 pts	RR 1.08 (0.98–1.19)

**Table 3 jcm-14-07982-t003:** Major bleeding definition.

Study	Major Bleeding Definition Used
Albabtain, 2021 [19]	NA
Alcalai, 2022 [12]	ISTH
Jones, 2021 [15]	ICH, major GI bleeding, hospitalization
Byrne, 2023 [6]	NA
Jaidka, 2018 [20]	NA
Jones, 2021 [15]	BARC > 2
Liang, 2022 [21]	TIMI
Mansouri, 2024 [22]	NA
Yao, 2025 [23]	ICH and GI bleeding
Youssef, 2022 [13]	BARC ≥ 2
Zhang, 2022 [24]	ISTH
RIVAWAR, 2025 [14]	ISTH

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
