# Peer review of "Anticoagulation Strategies for Left Ventricular Thrombus After Myocardial Infarction: A Review"

_jcm, 2025, doi:10.3390/jcm14227982_

Round 1

Reviewer 1 Report

Comments and Suggestions for Authors

MAJOR POINTS

The manuscript addresses an important clinical topic and provides a useful overview of current evidence on anticoagulation for left ventricular thrombus after myocardial infarction. However, there are several major aspects that require revision.

- While the manuscript defines itself as a narrative review, some parts are written with the tone of a meta-analysis, especially when reporting risk ratios and pooled estimates. This creates ambiguity in the reader’s perception of the methodology. It would be important to clarify that these numbers come from previously published studies and should not be interpreted as results of an original pooled analysis performed by the authors. A clearer methodological positioning would improve scientific transparency.

- The discussion tends at times to overstate the strength of the evidence supporting DOACs. Although the data are promising, most randomized trials are small and not powered for hard endpoints such as stroke or mortality. The conclusions should reflect a slightly more cautious tone, emphasizing that further large randomized studies are needed before DOACs can be considered equivalent or superior to VKAs in all scenarios.

- The section on therapy management would benefit from a more clinically oriented perspective. Since the greatest challenge in LVT treatment is balancing embolic risk and bleeding risk—especially in patients undergoing PCI with antiplatelet therapy—the reader would benefit from a clearer narrative on how treatment decisions can be individualized in real practice.

- The manuscript should discuss non-anticoagulation strategy to prevent cardio-embolic events (10.3390/biomedicines13030660).

MINOR POINTS

There are also minor issues concerning style, clarity, and language. In some sections, abbreviations are not used consistently, and punctuation could be improved to make long sentences more readable.     

- There are multiple instances where a sentence ends and the next sentence begins without a space (e.g., “…interactions.This review…”). 

- Several introductory words or phrases (e.g., However, Therefore, Moreover, In addition, Notably) are missing the comma that should follow them. Example: However warfarin remains… → However, warfarin remains…

- Section header is written as “2.methods”. Correct to “2. Methods” for consistency with other sections.

- Some sentences are notably long and would benefit from being split into two for clarity and smoother reading. This occurs mainly in the Introduction and Discussion. Consider simplifying by breaking at natural conceptual pauses.

- Ensure that each abbreviation (LVT, MI, DOAC, VKA) is defined at first use only, and then used consistently. In a few sections, abbreviations are re-introduced or spelled out again unnecessarily.

- Terms such as “follow up” and “follow-up” appear inconsistently. Use: follow-up (noun/adjective) → “follow-up imaging” follow up (verb) → “to follow up the patient”

- Minor plural/singular agreement issues. Example: guideline recommend → guidelines recommend, this data suggest → these data suggest or this evidence suggests (preferable in scientific writing)

- Both “thrombus resolution” and “LVT resolution” are used interchangeably. Either is fine, but choose one phrasing and use it consistently to improve flow.

- While rare, avoid informal contractions (e.g., “there’s”). Scientific English should always use the full form (“there is”).

Author Response

We thank Reviewer 1 for their careful reading and insightful comments, which have helped us improve the clarity, methodological transparency, and clinical relevance of our review. We have revised the manuscript accordingly. Specific responses to each comment are detailed below.

Comments and Suggestions for Authors

MAJOR POINTS

The manuscript addresses an important clinical topic and provides a useful overview of current evidence on anticoagulation for loo after myocardial infarction. However, there are several major aspects that require revision.

- While the manuscript defines itself as a narrative review, some parts are written with the tone of a meta-analysis, especially when reporting risk ratios and pooled estimates. This creates ambiguity in the reader’s perception of the methodology. It would be important to clarify that these numbers come from previously published studies and should not be interpreted as results of an original pooled analysis performed by the authors. A clearer methodological positioning would improve scientific transparency.

We appreciate this observation. The Methods section has been revised to explicitly clarify that all quantitative data (risk ratios, confidence intervals, and pooled estimates) were derived from previously published meta-analyses and not calculated by the authors. A clarifying sentence was added:

All pooled risk ratios reported in this review are derived from previously published analyses, and should not be interpreted as results of an original pooled analysis performed.

- The discussion tends at times to overstate the strength of the evidence supporting DOACs. Although the data are promising, most randomized trials are small and not powered for hard endpoints such as stroke or mortality. The conclusions should reflect a slightly more cautious tone, emphasizing that further large randomized studies are needed before DOACs can be considered equivalent or superior to VKAs in all scenarios.

We agree and have revised the Abstract, Discussion, and Conclusions to adopt a more balanced and cautious tone. The revised text now emphasizes that most available trials are small, underpowered, and hypothesis-generating. Statements such as “DOACs are at least as effective” were changed to “DOACs may be comparable,” and categorical claims were replaced with “suggest” or “indicate.” The Conclusions now read:

This review consolidates the available evidence comparing VKAs and DOACs in the treatment of LVT following MI. Available evidence - primarily from observational cohorts and small randomized trials - suggests that DOACs may be comparable to warfarin in achieving thrombus resolution and preventing embolic events, with a possible reduction in major bleeding. However, most studies are limited by modest sample sizes, short follow-up, and absence of hard clinical endpoints such as stroke or mortality. While these findings are encouraging, they should be interpreted with caution. The randomized data to date are insufficient to establish the non-inferiority or superiority of DOACs for all patient subgroups. Until large, adequately powered trials are completed, VKAs remain the reference standard, and DOACs should be considered selectively — particularly when INR control is unstable or contraindications to VKAs exist.

- The section on therapy management would benefit from a more clinically oriented perspective. Since the greatest challenge in LVT treatment is balancing embolic risk and bleeding risk—especially in patients undergoing PCI with antiplatelet therapy—the reader would benefit from a clearer narrative on how treatment decisions can be individualized in real practice.

We appreciate this suggestion. To strengthen clinical applicability, we have expanded the Concomitant Antiplatelet Therapy section by adding the following paragraph:

“In clinical practice, the optimal regimen should therefore be individualized by weighing embolic and bleeding risks. Patients with large anterior infarctions or severely reduced LVEF may benefit from a longer duration of anticoagulation, whereas those with high bleeding risk or recent PCI often require earlier de-escalation to dual therapy. Risk assessment tools such as HAS-BLED or PRECISE-DAPT can assist in estimating bleeding risk, while imaging follow-up at 3 months allows reassessment of embolic risk once thrombus resolution is confirmed. This pragmatic, patient-centered approach reflects the nuanced decision-making required in managing LVT after MI rather than a uniform protocol for all patients.”

This addition integrates individualized, real-world considerations and directly addresses the reviewer’s point.

- The manuscript should discuss non-anticoagulation strategy to prevent cardio-embolic events (10.3390/biomedicines13030660).

We have incorporated a dedicated paragraph near the end of the Discussion highlighting non-anticoagulant strategies, including early reperfusion, ventricular unloading, neurohormonal blockade, and inflammation modulation.

Additionally, antiplatelet therapy optimization, tight control of inflammatory and metabolic risk factors, and structured cardiac rehabilitation contribute to lowering thrombotic potential. Emerging approaches targeting myocardial inflammation and prothrombotic signaling, as reviewed by Sgarra et al [35], illustrate how comprehensive post-MI management may complement anticoagulation in mitigating embolic risk.

The following reference has been added and discussed, the reference was added:

Sgarra L, Desantis V, Matteucci A, Caccavo VP, Troisi F, Di Monaco A, Mangini F, Katsouras G, Guaricci AI, Dadamo ML, Fortunato F, Nacci C, Potenza MA, Montagnani M, Grimaldi M. Non-Anticoagulation Strategies Aimed at Primary Stroke Prevention in Nascent Atrial Fibrillation. Biomedicines. 2025 Mar 7;13(3):660.

MINOR POINTS

There are also minor issues concerning style, clarity, and language. In some sections, abbreviations are not used consistently, and punctuation could be improved to make long sentences more readable.     

- There are multiple instances where a sentence ends and the next sentence begins without a space (e.g., “…interactions.This review…”). 

All spacing errors following periods and punctuation marks have been corrected throughout the manuscript.

- Several introductory words or phrases (e.g., However, Therefore, Moreover, In addition, Notably) are missing the comma that should follow them. Example: However warfarin remains… → However, warfarin remains…

We reviewed and corrected all such instances. Introductory words and transitions are now consistently followed by commas (e.g., “However, warfarin remains…”).

- Section header is written as “2.methods”. Correct to “2. Methods” for consistency with other sections.

The formatting of the section heading has been corrected to ‘2. Methods’ for consistency with the other section titles.

- Some sentences are notably long and would benefit from being split into two for clarity and smoother reading. This occurs mainly in the Introduction and Discussion. Consider simplifying by breaking at natural conceptual pauses.

We identified and revised multiple long or compound sentences in the Introduction and Discussion sections. These were divided at natural conceptual pauses to improve readability and flow.

- Ensure that each abbreviation (LVT, MI, DOAC, VKA) is defined at first use only, and then used consistently. In a few sections, abbreviations are re-introduced or spelled out again unnecessarily.

All abbreviations were reviewed and standardized. Each term is now defined at its first appearance and used consistently thereafter without redefinition.

- Terms such as “follow up” and “follow-up” appear inconsistently. Use: follow-up (noun/adjective) → “follow-up imaging” follow up (verb) → “to follow up the patient”

We harmonized all occurrences according to proper usage: “follow-up” when used as a noun/adjective, and “follow up” when used as a verb.

- Minor plural/singular agreement issues. Example: guideline recommend → guidelines recommend, this data suggest → these data suggest or this evidence suggests (preferable in scientific writing)

Grammatical inconsistencies were corrected (e.g., “guidelines recommend,” “these data suggest,” “this evidence suggests”), ensuring proper scientific language.

- Both “thrombus resolution” and “LVT resolution” are used interchangeably. Either is fine, but choose one phrasing and use it consistently to improve flow.

We standardized terminology throughout the manuscript to use “thrombus resolution” consistently.

- While rare, avoid informal contractions (e.g., “there’s”). Scientific English should always use the full form (“there is”).

All informal contractions have been replaced with formal equivalents (e.g., “there is,” “it has”) to maintain professional tone.

Reviewer 2 Report

Comments and Suggestions for Authors

Dear Authors,

Your review paper is of current interest, well-written, and structured. However, in my opinion, a few minor revisions are required to improve its quality:

  • Although it is a narrative review, it would be helpful to report how many studies were initially identified, screened, excluded, and included. A flow summary or simple diagram illustrating this process would be valuable.
  • The discussion section references only the 2017 ESC guideline for STEMI. However, the 2023 ESC Guidelines for ACS address antithrombotic therapy—including both warfarin and NOACs in the context of left-ventricular thrombus treatment—though without a clear statement on optimal duration or combination. In this regard, it would be of interest to insert a short paragraph discussing these newer guideline recommendations.
  • Line 94 – spelling error: the capital letter was missed in the Methods section.
  • Abbreviation explanations used in the table must be inserted beneath the table.

Author Response

REVIEWER 2

Dear Authors,

Your review paper is of current interest, well-written, and structured. However, in my opinion, a few minor revisions are required to improve its quality:

  • Although it is a narrative review, it would be helpful to report how many studies were initially identified, screened, excluded, and included. A flow summary or simple diagram illustrating this process would be valuable.

We thank the reviewer for this suggestion. While our work is a narrative review and therefore not required to adhere to the full PRISMA protocol, we agree that presenting the literature selection process improves transparency and readability. Accordingly, we have added a simplified flow chart (new Figure 1) summarizing the number of studies identified, screened, excluded, and ultimately included in the qualitative synthesis. This addition enhances methodological clarity while maintaining the narrative nature of the review.

  • The discussion section references only the 2017 ESC guideline for STEMI. However, the 2023 ESC Guidelines for ACS address antithrombotic therapy—including both warfarin and NOACs in the context of left-ventricular thrombus treatment—though without a clear statement on optimal duration or combination. In this regard, it would be of interest to insert a short paragraph discussing these newer guideline recommendations.

We fully agree and have revised the Discussion section to include the 2023 ESC Guidelines for the Management of Acute Coronary Syndromes. The new paragraph highlights that these updated guidelines mention both VKAs and DOACs as potential options for confirmed left-ventricular thrombus (Class IIa, Level C) but do not specify an optimal agent, duration, or combination strategy. This addition aligns the manuscript with the most recent European recommendations and emphasizes the persisting evidence gap in this area.

  • Line 94 – spelling error: the capital letter was missed in the Methods section.

Thank you for catching this. The typographical error has been corrected; the Methods section now begins with a capital letter as appropriate.

  • Abbreviation explanations used in the table must be inserted beneath the table.

Thank you, this was added to the table:

ACS – acute coronary syndrome, A – aspirin, A+C – aspirin plus clopidogrel, C – clopidogrel, CMR – cardiac magnetic resonance imaging, DAPT – dual antiplatelet therapy, DOAC – direct oral anticoagulant, Edox – edoxaban, Echo – echocardiography, INR – international normalized ratio,LVT – left ventricular thrombus, MI – myocardial infarction, mo – month(s), NA – not available / not applicable, NR – not reported, PCI – percutaneous coronary intervention, RCT – randomized controlled trial, Riva – rivaroxaban, Apix – apixaban, Dabig – dabigatran, SAPT – single antiplatelet therapy, STEMI – ST-segment elevation myocardial infarction, TTE – transthoracic echocardiography, VKA – vitamin K antagonist, y – year(s)

Reviewer 3 Report

Comments and Suggestions for Authors

My congratulations to the authors for their interesting review; despite its quality, I’ve found some aspects that need to be corrected:

The manuscript would be strengthened by the inclusion of recent data addressing the pharmacoeconomic impact of DOACs and outcomes focused on patient experience and quality of life.

Although the Results and Discussion sections are generally well organized, they contain some repetition. For example, lines 193–210 reiterate information already presented earlier in the Results. These sections could be streamlined by consolidating the trial descriptions into a single comparative paragraph and relocating detailed study characteristics to a summary table or supplementary appendix.

The Methods section outlines a literature search strategy but lacks important methodological details, such as: the specific databases searched (e.g., MEDLINE, Scopus); the precise time frame of the search (currently noted only as “through August 2025”). The inclusion and exclusion criteria (it is unclear whether studies without comparator arms were omitted); the study selection process (no PRISMA-style flow diagram is provided).

Indeed authors are encouraged to include real world data of DOACs safety (doi: 10.23736/S2724-5683.24.06546-3) to their discussion in order to enrich their clinical context

Moreover no figures are provided, even though the topic lends itself well to visual representation—such as a flowchart outlining treatment strategies or a schematic depiction of LVT formation. The lack of visual elements reduces the paper’s accessibility, particularly for readers less familiar with this clinical area. Authors should expand this aspect.

Author Response

REVIEWER 3

My congratulations to the authors for their interesting review; despite its quality, I’ve found some aspects that need to be corrected:

The manuscript would be strengthened by the inclusion of recent data addressing the pharmacoeconomic impact of DOACs and outcomes focused on patient experience and quality of life.

We fully agree that the pharmacoeconomic implications and patient-centered outcomes of DOAC therapy represent key aspects of real-world clinical decision-making. Accordingly, we have expanded the Discussion section to include recent evidence regarding the cost-effectiveness of DOACs compared with warfarin and their association with improved patient satisfaction, treatment adherence, and quality of life. The updated paragraph cites the most recent data available from large observational and modeling studies and highlights the need for further prospective health-economic evaluations in this population.

Although the Results and Discussion sections are generally well organized, they contain some repetition. For example, lines 193–210 reiterate information already presented earlier in the Results. These sections could be streamlined by consolidating the trial descriptions into a single comparative paragraph and relocating detailed study characteristics to a summary table or supplementary appendix.

We thank the reviewer for this thoughtful and constructive observation. We carefully revisited both the Results and Discussion sections to identify and reduce areas of repetition. While preserving essential content, we streamlined the narrative to ensure smoother flow and avoid redundancy. We believe these modifications have improved the overall coherence and presentation of the manuscript, and we are grateful to the reviewer for highlighting this point.

The Methods section outlines a literature search strategy but lacks important methodological details, such as: the specific databases searched (e.g., MEDLINE, Scopus); the precise time frame of the search (currently noted only as “through August 2025”). The inclusion and exclusion criteria (it is unclear whether studies without comparator arms were omitted); the study selection process (no PRISMA-style flow diagram is provided).

We have expanded the Methods section to specify the databases searched (PubMed, Embase, and Cochrane Library), the time frame (“from database inception through August 2025”), and explicit inclusion/exclusion criteria (excluding studies lacking comparator arms or not involving post-MI LVT populations). To improve transparency, a simplified flow diagram (Figure 1) has been added to depict identification, screening, exclusion, and inclusion steps. These additions enhance methodological clarity while maintaining the narrative nature of the review.

Indeed authors are encouraged to include real world data of DOACs safety (doi: 10.23736/S2724-5683.24.06546-3) to their discussion in order to enrich their clinical context

We have incorporated the cited real-world study into the Discussion and summarized its findings. The new sentence reads:

Furthermore, real-world safety data reinforce the clinical applicability of DOACs. A recent multicenter [40] reported that DOAC use in post-MI and cardiomyopathic populations was associated with low major bleeding rates and favorable net clinical benefit compared with VKAs. Such evidence strengthens confidence in the safety profile of DOACs beyond controlled trial settings.

  1. And added ref 40 - Lavalle C, Pierucci N, Mariani MV, et al. Italian Registry in the Setting of Atrial Fibrillation Ablation with Rivaroxaban - IRIS. Minerva Cardiol Angiol. 2024;72(6):625-637. doi:10.23736/S2724-5683.24.06546-3

Moreover no figures are provided, even though the topic lends itself well to visual representation—such as a flowchart outlining treatment strategies or a schematic depiction of LVT formation. The lack of visual elements reduces the paper’s accessibility, particularly for readers less familiar with this clinical area. Authors should expand this aspect.

We thank the reviewer for this valuable suggestion. In response, we have added new visual material to enhance the manuscript’s accessibility and educational value. Specifically, we created a schematic figure illustrating the pathophysiological mechanisms and treatment targets in left ventricular thrombus (new Figure 2). The illustration integrates the key components of Virchow’s triad -blood stasis, hypercoagulability, and endothelial injury - together with their corresponding therapeutic strategies. The figure provides readers with a concise visual summary of LVT formation and management, complementing the narrative discussion and making the manuscript more approachable to a broader audience.

Round 2

Reviewer 1 Report

Comments and Suggestions for Authors

No other suggestions 

Reviewer 3 Report

Comments and Suggestions for Authors

Authors must be congratulated for the revised version of the manuscript.